# Metabolic Syndrome Severity Score, Comparable to Serum Creatinine, Could Predict the Occurrence of End-Stage Kidney Disease in Patients with Antineutrophil Cytoplasmic Antibody-Associated Vasculitis

**DOI:** 10.3390/jcm10245744

**Published:** 2021-12-08

**Authors:** Pil Gyu Park, Jung Yoon Pyo, Sung Soo Ahn, Jason Jungsik Song, Yong-Beom Park, Ji Hye Huh, Sang-Won Lee

**Affiliations:** 1Division of Rheumatology, Department of Internal Medicine, Yonsei University College of Medicine, Seoul 03722, Korea; pilgyu.park@yuhs.ac (P.G.P.); jyp@yuhs.ac (J.Y.P.); saneth@yuhs.ac (S.S.A.); jsksong@yuhs.ac (J.J.S.); yongbpark@yuhs.ac (Y.-B.P.); 2Institute for Immunology and Immunological Diseases, Yonsei University College of Medicine, Seoul 03722, Korea; 3Division of Endocrinology and Metabolism, Department of Internal Medicine, Hallym University Sacred Heart Hospital, Anyang 14068, Gyeonggi-do, Korea

**Keywords:** antineutrophil cytoplasmic antibody, vasculitis, metabolic syndrome, severity score, end-stage kidney disease

## Abstract

This study investigated whether the metabolic syndrome (MetS) severity (MSSS) at diagnosis could predict poor outcomes during follow-up in antineutrophil cytoplasmic antibody (ANCA)-associated vasculitis (AAV) patients with MetS. The equation for the MSSS at diagnosis used in this study was developed and validated in Korean adults aged 20–59 years. The medical records of 261 patients with AAV were retrospectively reviewed, and finally, 36 AAV patients with MetS aged 20–59 years fulfilling the inclusion criteria were included in this study. All-cause mortality, relapse, end-stage kidney disease (ESKD), cerebrovascular accident, and cardiovascular disease were assessed as the poor outcomes of AAV. Their median age was 51.2 years and 36.1% were male. The MSSS was significantly correlated with age and serum albumin but not AAV-specific indices. Among the five poor outcomes, only ESKD showed a relatively significant area under the curve (area 0.696) in receiver operating characteristic curve analysis. In the multivariable Cox hazards model analysis, both serum creatinine (HR 3.033) and MSSS (HR = 2.221) were significantly associated with ESKD occurrence. When the cut-off of the MSSS for ESKD was set at 1.72, ESKD occurred more frequently in patients with MSSS ≥ 1.72 than in those with MSSS < 1.72 (75.0% versus 14.3%, *p* = 0.002). Furthermore, patients with MSSS ≥ 1.72 exhibited a significantly lower cumulative ESKD-free survival rate than those with MSSS < 1.72 (*p* = 0.001). MSSS at the time of AAV diagnosis independently predicted the occurrence of ESKD during follow-up in patients with AAV and MetS.

## 1. Introduction

The 2012 revised International Chapel Hill Consensus Conference Nomenclature of Vasculitides defined antineutrophil cytoplasmic antibody (ANCA)-associated vasculitis (AAV) as necrotising vasculitis with few or no immune deposits that mainly affects small vessels, such as capillaries, arterioles, and venules [1]. While AAV traditionally comprises three subtypes, including eosinophilic granulomatosis with polyangiitis (Churg–Strauss syndrome), granulomatosis with polyangiitis (Wegener’s granulomatosis), and microscopic polyangiitis [2], it has recently been categorised into three subtypes according to ANCA types as myeloperoxidase-ANCA, proteinase 3-ANCA, and ANCA-negative vasculitis [1,3].

ANCA vasculitis has relatively high mortality and complication rates compared to other vasculitides [4,5]. While many efforts have been made to develop an index to predict poor outcomes, an independent and clear index with clinical reliability is still lacking. Indices that predict these poor outcomes at AAV diagnosis may allow time to delay their progression and possible mortality. Therefore, efforts to discover these indices must be continued.

Metabolic syndrome (MetS) is a cluster of medical conditions that are associated with central obesity, hypertension, dyslipidaemia, and impaired glucose tolerance, which may also increase the risks of cardiovascular disease (CVD), cerebrovascular accident (CVA), and type 2 diabetes mellitus (T2DM) [6]. Regarding the link between MetS and AAV, MetS is both a cause and a result of chronic inflammation and AAV is a representative inflammatory disease [7,8]; thus, MetS may influence the prognosis of AAV. A previous study conducted in the Netherlands reported that MetS occurred more frequently in patients with AAV than in controls. In addition, AAV patients with MetS showed higher rates of relapse during the disease course [9]. Similarly, we recently reported a higher prevalence of MetS in patients with AAV than in healthy controls in Korea [10].

The MetS severity score (MSSS) is a new index reflecting the severity of MetS. Currently, MetS is diagnosed when three of the five components of the diagnostic classification criteria are fulfilled. However, the use of the conventional MetS diagnostic criteria (Adult Treatment Program III [ATP-III]) has been limited by the dichotomy of the current MetS definitions (i.e., present/absent) [11]. Based on the current diagnostic criteria, it is difficult to assess the severity of metabolic risk for individuals, particularly in patients who have already had MetS. Therefore, the MSSS was developed to reflect subtle differences in MetS severity [12,13]. Our research group recently developed and validated an MSSS for the general Korean population [14,15]. Unlike the presence of MetS, to date, no study has assessed the clinical impact of the MSSS on poor outcomes in patients with AAV. Hence, this study investigated whether the MSSS at the time of AAV diagnosis could predict poor outcomes during follow-up in AAV patients with MetS.

## 2. Materials and Methods

### 2.1. Patients

The medical records of 261 patients with AAV were retrospectively reviewed. All patients were enrolled in the Severance Hospital ANCA-associated VasculitidEs (SHAVE) cohort, a prospective and observational cohort of AAV patients, beginning in November 2016. The inclusion criteria were: (i) AAV fulfilling both the classification algorithm for AAV and polyarteritis nodosa proposed by the European Medicines Agency in 2007 and the revised nomenclature of vasculitides suggested by the Chapel Hill Conference Consensus in 2012 [1,2]; (ii) AAV first classified in the Rheumatology Division of Yonsei University College of Medicine, Severance Hospital; (iii) medical records sufficiently complete to provide clinical and laboratory data for the assessment of AAV-specific indices at AAV, such as the Birmingham vasculitis activity score (BVAS) and five-factor score (FFS) [16,17]; (iv) medical records describing the items to which the diagnostic classification criteria for metabolic syndrome could be applied at the time of AAV diagnosis; (v) MetS diagnosed at the time of AAV; and (vi) AAV patient age at the time of diagnosis between 20 and 59 years as an MSSS equation for patients over 60 years of age is lacking. The exclusion criteria were: (i) follow-up duration <3 months from the time of AAV diagnosis; (ii) immunosuppressive drugs for AAV treatment had been ever administered before a diagnosis of AAV; and (iii) serious medical conditions, such as concurrent malignancies, infectious diseases requiring hospitalisation, and other systemic vasculitides.

Of 261 patients with AAV, 68 were excluded due to a lack of information on the five items assessed for MetS diagnosis. Of the remaining 193 patients, 99 were excluded because they were not diagnosed with MetS. Among the remaining 94 patients with MetS, 58 were excluded for age <19 or >60 years. As a result, this study included and analysed 36 AAV patients with MetS aged 20–59 years at the time of AAV diagnosis (Figure 1). This study was approved by the Institutional Review Board (IRB) of Severance Hospital (Seoul, Korea, IRB No. 4-2020-1071) and conducted according to the principles of the Declaration of Helsinki. Given the retrospective design of the study and the use of anonymised patient data, the requirement for written informed consent was waived.

### 2.2. Definition of MetS

According to the modified NCEP ATP-III criteria, MetS can be diagnosed when more than three of five MetS components for Asians are fulfilled [11,18]. The five MetS components are as follows: (i) abdominal obesity (waist circumference (WC) ≥ 90 cm in men and ≥80 cm in women); (ii) high blood pressure (systolic blood pressure (SBP) ≥ 130 mm Hg, diastolic blood pressure ≥ 85 mm Hg, or treatment with antihypertensive agents); (iii) hypertriglyceridaemia (serum triglyceride (TG) concentration ≥ 150 mg/dL); (iv) low high-density lipoprotein (HDL)-cholesterol level (serum HDL-C < 40 mg/dL for men or <50 mg/dL for women), and (v) high fasting glucose (fasting plasma glucose (FPG) ≥ 100 mg/dL or a previous diagnosis of T2DM).

### 2.3. MSSS Equation

The equation for the MSSS used in this study was developed and validated in Korean adults [14]: The MSSS equation for men (20–59 years) = −8.5245 + 0.0156 × FPG + 0.0089 × SBP + 0.0371 × WC − 0.0182 × HDL-cholesterol + 0.7913 × ln (TG); and the MSSS equation for women (20–59 years) = −8.4480 + 0.0223 × FPG + 0.0115 × SBP + 0.0403 × WC − 0.0155 × HDL-cholesterol + 0.6704 × ln (TG).

### 2.4. Variables

The demographic data at the time of AAV included age, sex, and body mass index (BMI). AAV subtypes, ANCA type, AAV-specific indices (BVAS and FFS), and laboratory data were also collected. The erythrocyte sedimentation rate (ESR) and C-reactive protein (CRP) levels were assessed as acute-phase reactants. Blood samples for the measurement of serum glucose and total cholesterol levels were obtained after an overnight fast. The MSSS was calculated using the equation for the MSSS in men and women [14].

During follow-up, poor outcomes were evaluated, and the follow-up duration based on each poor outcome was calculated. In this study, all-cause mortality, relapse, end-stage kidney disease (ESKD), CVA, and CVD were defined as poor outcomes in patients with AAV. Relapse, ESKD, CVA, and CVD were defined as described previously [5]. The follow-up duration based on all-cause mortality was defined as the period between AAV diagnosis and the last visit for surviving patients and as the period between AAV diagnosis and death for deceased patients. For patients with a poor outcome, the follow-up duration based on each poor outcome was defined as the period between AAV diagnosis and the occurrence of each poor outcome. Conversely, for patients with no poor outcome, the follow-up duration was defined as the period between AAV diagnosis and the last visit. The numbers of patients who received glucocorticoids and immunosuppressive drugs for AAV treatment during follow-up were also collected.

### 2.5. Statistical Analyses

All statistical analyses were performed using IBM SPSS Statistics for Windows version 25.0 (IBM Corp., Armonk, NY, USA). Continuous and categorical variables are expressed as medians with interquartile ranges and numbers (percentages), respectively. The correlation coefficient (r) between the two variables was obtained using the Pearson correlation analysis. The poor outcome as a dependent variable was determined to have a *p*-value of <0.1 in receiver operating characteristic (ROC) curve analysis. The multivariable Cox hazard model using variables with a *p*-value of <0.1 in the univariable Cox hazard model was used to obtain the hazard ratios (HRs) during the follow-up duration. The optimal cut-off was extrapolated by performing ROC curve analysis to identify the value with the maximum sum of sensitivity and specificity. The relative risk (RR) of the cut-off for high AAV activity was analysed using contingency tables and chi-square tests. Comparisons of the cumulative survival rates between the two groups were performed using Kaplan–Meier survival analysis with log-rank tests. Statistical significance was set at *p* < 0.05.

## 3. Results

### 3.1. Characteristics of AAV Patients with MetS

The characteristics of the 36 patients at the time of AAV diagnosis are summarised in Table 1. Their median age was 51.2 years and 36.1% were male. The median values of BVAS, FFS, ESR, and CRP were 12.5, 1.0, 62.0 mm/hr, and 8.5 mg/L, respectively. The median MSSS was 1.1. During follow-up, one patient (2.8%) died, and 20 patients (55.6%) experienced a relapse. ESKD, CVA, and CVD occurred in 10 (27.8%), 2 (5.6%), and 5 (13.9%) patients, respectively.

### 3.2. Correlation Analysis

Among the continuous variables at the time of AAV, age (r = 0.340, *p* = 0.042) was significantly correlated with the MSSS, whereas serum albumin (r = −0.363, *p* = 0.030) was inversely correlated with the MSSS. However, the MSSS was not significantly correlated with BVAS, FFS, or acute-phase reactants. Therefore, the MSSS did not reflect the simultaneous activity nor the inflammatory burden in AAV patients with MetS (Appendix A).

### 3.3. Determination of the Target Poor Outcome

Among the five poor outcomes, only ESKD showed relative significance in the area under the curve in the ROC analysis (area 0.696, *p* = 0.072). Therefore, this study defined ESKD as the target poor outcome as mentioned in the methods section (Figure 2).

### 3.4. Cox Hazards Model Analyses for the Occurrence of ESKD

In the univariable Cox hazard model analysis, BMI (HR = 0.779), BVAS (HR = 1.114), FFS (HR = 2.716), haemoglobin (HR = 0.665), blood urea nitrogen (HR = 1.023), serum creatinine (HR = 2.508), ESR (HR = 1.019), and MSSS (HR = 1.399) at the time of AAV were significantly associated with the occurrence of ESKD during follow-up. Since serum creatinine and creatinine clearance are directly affected by BMI, BMI was excluded from the multivariable Cox analysis [19]. In the multivariable analysis, both serum creatinine (HR 3.713, 95% confidence interval [CI] 1.560–8.838) and MSSS (HR = 1.971, 95% CI 1.071–3.630) were significantly associated with the occurrence of ESKD (Table 2). Therefore, the MSSS at the time of AAV was independently associated with the occurrence of ESKD during follow-up in patients with AAV and MetS.

### 3.5. Optimal MSSS Cut-Off and RR for ESKD

The sensitivity and specificity of an MSSS cut-off value of ≥1.72 for predicting the occurrence of ESKD were 60.6% and 92.3%, respectively. When we divided patients into two groups based on this threshold, 8 of 36 patients had an MSSS ≥ 1.72. ESKD occurred more frequently in patients with MSSS ≥ 1.72 than in those with MSSS < 1.72 (75.0% versus 14.3%, *p* = 0.002). Furthermore, patients with MSSS ≥1.72 had a significantly higher risk of ESKD than those with MSSS < 1.72 (RR = 18.000, 95% CI 2.642–122.617) (Figure 3).

### 3.6. Comparisons between the Two Groups According to MSSS ≥ 1.72

Among poor outcomes, more patients with MSSS ≥ 1.72 than those with MSSS < 1.72 (75.0% versus 14.3%, *p* = 0.002) showed progression to ESKD. Meanwhile, there was no difference in the history of immunosuppressive drug use between the two groups (Appendix A).

### 3.7. Comparisons of Cumulative ESKD-Free Survival Rates between the Two Groups According to MSSS ≥ 1.72

Patients with MSSS ≥ 1.72 exhibited a significantly lower cumulative ESKD-free survival rate than those with MSSS < 1.72 (*p* = 0.001) (Figure 4).

## 4. Discussion

In this study, we first demonstrated that the MSSS at the time of AAV diagnosis, which is a continuous estimate of MetS severity, predicted the occurrence of ESKD during follow-up in AAV patients with MetS. We assessed whether the MSSS only played a role as an indirect predictor of ESKD or directly affected the occurrence of ESKD in patients with AAV and MetS. First, apart from AAV, MetS may induce and accelerate ESKD occurrence in the general population [20], and it was significantly associated with the occurrence of ESKD in patients with chronic stage 3 and 4 kidney disease [21]. In addition, T2DM or impaired fasting glucose (IFG) is a risk factor for ESKD [22], while hypertension also increases the incidence of ESKD in individuals without kidney diseases [23]. Therefore, it can be assumed that MetS itself, which is composed of T2DM, IFG, and hypertension, may reduce kidney function and ultimately accelerate the progression to ESKD in individuals with MetS.

In terms of AAV, we conducted Kaplan–Meier survival analysis and compared the cumulative rates of each poor outcome among all-cause mortality, relapse, ESKD, CVA, and CVD between AAV patients with MetS and those without MetS. We found there was a significant difference in only the cumulative ESKD-free survival rates between the two groups (*p* = 0.030). (Appendix A). Therefore, the risk of ESKD caused by MetS may be added to the risk of ESKD associated with AAV itself. These results might suggest a rationale for the need to confirm the presence of MetS in patients with AAV and to control metabolic profiles, such as high blood pressure, high blood glucose, and dyslipidaemia, in addition to controlling the activity of vasculitis in AAV patients with MetS.

The univariable Cox analysis to determine which of the five variables constituting the MSSS contributed to the association between MSSS and the occurrence of ESKD showed a significant HR for FPG (HR = 1.009), in addition to BVAS (HR = 1.114), FFS (HR = 2.726), haemoglobin (HR = 0.665), blood urea nitrogen (HR = 1.023), serum creatinine (HR = 2.508), and serum albumin (HR = 0.555). The multivariable Cox analysis including the factors statistically significant in the univariate analysis showed that both FPG (HR = 1.014, 95% CI 1.000–1.029) and serum creatinine (HR = 2.778, 95% CI 1.473–5.236) levels were significantly and independently associated with the occurrence of ESKD, similar to the MSSS (Appendix A).

The FPG cut-off (≥140.0 mg/dL) for predicting ESKD was obtained from ROC curve analysis (sensitivity 50.0%, specificity 96.2%, area 0.760, *p* = 0.017). When we divided the patients into two groups based on the FPG cut-off of ≥140.0 mg/dL, ESKD was identified more frequently in AAV patients with FPG ≥ 140.0 mg/dL than in those with FPG < 140.0 mg/dL (83.3% versus 16.7%, *p* = 0.003). AAV patients with FPG ≥ 140.0 mg/dL exhibited a significantly higher risk of ESKD than those with FPG < 140.0 mg/dL (RR 25.000, 95% CI 2.380–262.653) (Appendix A). In addition, patients with FPG ≥ 140.0 mg/dL exhibited a significantly lower cumulative ESKD-free survival rate than those with FPG < 140.0 mg/dL (*p* < 0.001) (Appendix A). Therefore, similar to the MSSS, FPG contributed to the association between MSSS and ESKD and FPG showed potential as a predictor of ESKD in AAV patients.

Our group previously demonstrated that continuous MSSS had a high predictive ability for the development of CVD based on data from the Korean population [14]. Therefore, we hypothesised that MetS severity assessed by the MSSS could also be used to predict the microvascular outcomes of patients with AAV and MetS. We observed that AAV patients with a severe degree of MetS risk, as assessed by MSSS, were at a higher risk of ESKD. This result may not be surprising as MetS is already considered a risk factor for chronic kidney disease [24]. However, the traditional MetS criteria are limited by identifying a risk for chronic kidney disease, only when a person exhibits abnormalities beyond the cut-offs for three of the components. Moreover, the binary nature of the traditional MetS criteria has a limitation on representing the metabolic risk changes over time. However, continuous MSSS can be used to follow up on individual changes in metabolic risk over time. As the risk of metabolic abnormalities gradually increases over time in patients with AAV, tracking the change in metabolic risk using MSSS can help to identify the timing for attentive intervention to prevent vascular complications. Therefore, the MSSS might be more beneficial than the traditional ATP-III criteria in predicting vascular complications, such as ESKD, and assessing changes in ESKD risk over time in patients with AAV.

The present study calculated the MSSS using the equation for adults with MetS aged 20–59 years to investigate the association between MetS severity and poor AAV outcome. For this reason, owing to the age limitation in this equation, this study analysed only 36 of 94 patients with AAV and MetS [14]. The fact that the oldest patient included in our cohort was 79 years of age, and considering the high prevalence of MetS in older patients, the development of an equation for MSSS for patients up to 80 years would be valuable. It would expand its ability to predict health outcomes in all 94 patients with AAV and MetS. However, there are several challenges in developing an equation for MSSS in individuals over 60 years of age. First, as older people are likely to have comorbidities and take various medications, it may influence the MSSS. Nevertheless, as older people are increasing, the development of an equation for calculating MSSS in individuals up to 80 years of age could lead to a good prognosis by actively treating MetS, owing to the predictive power of the MSSS for poor prognoses.

The most reliable predictors of progression to ESKD in AAV patients are kidney histopathological findings, in particular, those obtained using the using the renal risk scoring systems, such as Berden classification, Mayo Clinic/Renal Pathology Society Chronicity Score, and ANCA renal risk score (APRS) [25,26,27]. However, in this study, the predictability of progression to ESKD could not be calculated using the kidney risk scoring system for several reasons. First, since the purpose of this study was to investigate the clinical implications of MSSS in AAV patients with MetS, not general AAV patients, 68 patients without information on the presence or absence of MetS were excluded. Therefore, it seems impossible to simply compare the abilities to predict ESKD between kidney biopsy results and MSSS in this study. Second, for patients whose kidney vasculitis was not clear at AAV diagnosis, it may be difficult to compare the abilities to predict ESKD in AAV patients between the MSSS calculated at AAV diagnosis and the kidney biopsy results performed a considerable time after diagnosis. Third, despite the clear clinical manifestations of kidney vasculitis at AAV diagnosis, there were patients without kidney biopsy results because the diagnosis was based on biopsy results performed in other organs. Fourth, despite the clear clinical manifestations of kidney vasculitis at AAV diagnosis based on the criteria proposed by the European Medicines Agency in 2007, there were patients without kidney biopsy results because they rejected kidney biopsy or had medical conditions where a kidney biopsy was contraindicated. Lastly, it would be an advantage of this study to develop a new index to replace kidney biopsy results in patients with confirmed MetS.

For the first time, this study demonstrated that the MSSS at the time of AAV diagnosis was independently and significantly associated with the occurrence of ESKD during follow-up in AAV patients with MetS. This study also provided a method to determine the optimal MSSS cut-off to predict the occurrence of ESKD in patients with AAV. As the MSSS equation used in this study is limited to an Asian population, the optimal MSSS cut-off value for predicting ESKD is limited to the Asian population as well. Thus, a confounding effect of racial or regional differences can be ignored. The results of this study also emphasised the need to evaluate and manage the inflammatory burden of MetS components, particularly T2DM or IFG and hypertension, to reduce the incidence of ESKD in AAV patients with MetS. Our findings provide evidence of the potential utility of MSSS in clinical practice to identify patients with AAV at increased risk of ESKD.

This retrospective study has several limitations. A limited number of patients were included owing to the number of patients with available information recorded in the medical records for diagnosing MetS. Additionally, the median age, at the time of AAV diagnosis, was 61 years in the SHAVE cohort. Since the equation for calculating MSSS in individuals over 60 years of age has not yet been developed, this study included a very limited number of patients. Nevertheless, we are confident that the present study presents clinically significant implications in two respects. One is that in terms of the ethnic difference, the AAV cohort investigated in this study is the only and largest cohort in Korea that has accumulated high-reliability clinical data according to the protocol. Therefore, it is practically impossible to include additional patients through multi-institutional participation in a short period of time. The other is that given that there has been no study evaluating the clinical significance of MSSS in Korean patients with AAV, it is thought that the results of this study also have clinical significance in terms of the pilot study design. Additionally, we will try to formally propose the development of an MSSS formula that can be applied to patients over 60 years of age to the relevant society. In addition, we analysed only MetS accompanied by AAV at the time of AAV diagnosis; MetS that occurred during follow-up was not investigated. Therefore, the clinical implications of MetS that occurred during follow-up were not assessed. A prospective study with a larger number of AAV patients with MetS and regular MetS assessment during follow-up is needed to validate the results of this study and to provide novel additional information on the impact of MSSS on the disease course of AAV.

## 5. Conclusions

MSSS at the time of AAV diagnosis independently predicted the occurrence of ESKD during follow-up in patients with AAV and MetS. After classifying AAV patients with MetS, physicians should determine the MSSS cut-off for the occurrence of ESKD using the MSSS equation appropriate to their ethnicity and need to pay careful attention to manage MetS in patients at high risk of ESKD. Furthermore, long-term follow-up studies are warranted to generalise the utility of the MSSS in predicting various types of health outcomes in patients with AAV.

## Figures and Tables

**Figure 1 jcm-10-05744-f001:**
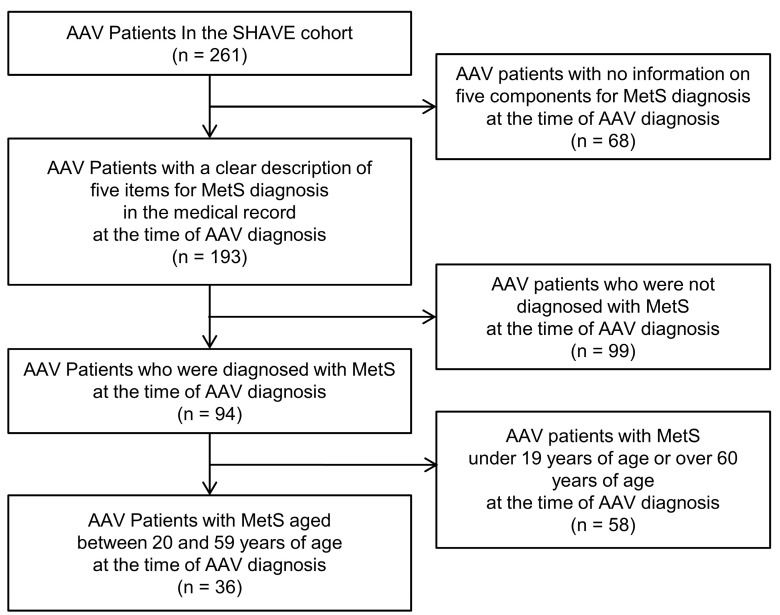
Algorithm for selection of the study subjects. AAV, ANCA-associated vasculitis; ANCA, antineutrophil cytoplasmic antibody; SHAVE, Severance Hospital ANCA-associated vasculitis; MetS, metabolic syndrome.

**Figure 2 jcm-10-05744-f002:**
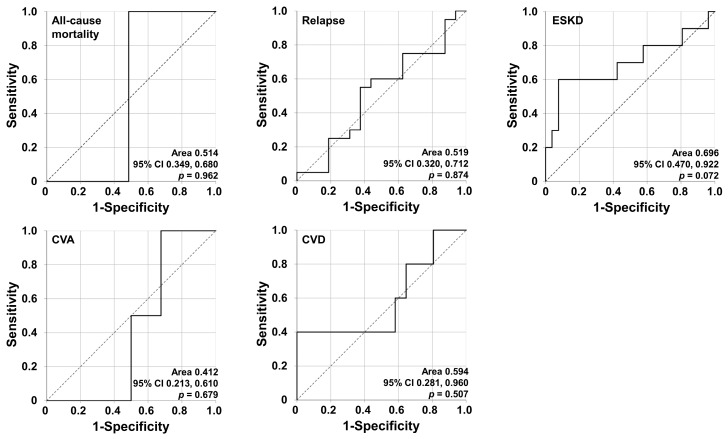
Determining the target poor outcome. ESKD was defined as the target poor outcome as it showed a relatively significant area under the curve in the ROC analysis. ESKD, end-stage kidney disease; CVA, cerebrovascular accident; CVD, cardiovascular disease; CI, confidence interval; ROC, receiver operating characteristic.

**Figure 3 jcm-10-05744-f003:**
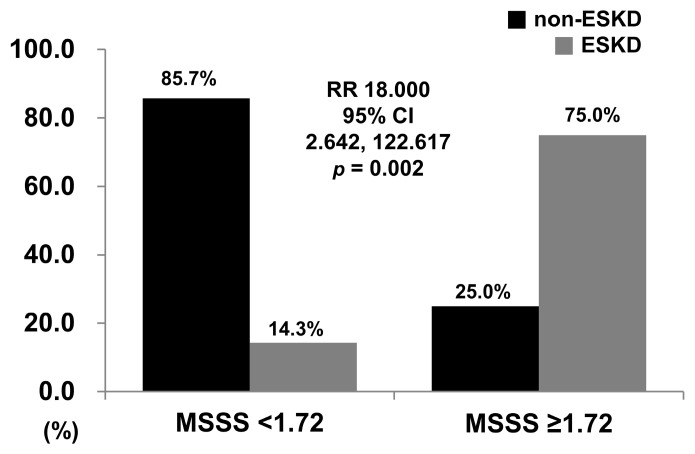
Relative risk of ESKD. Patients with an MSSS ≥ 1.72 showed a significantly higher risk of ESKD than patients with an MSSS < 1.72. MSSS, metabolic syndrome severity score; ESKD, end-stage kidney disease; RR, relative risk; CI, confidence interval.

**Figure 4 jcm-10-05744-f004:**
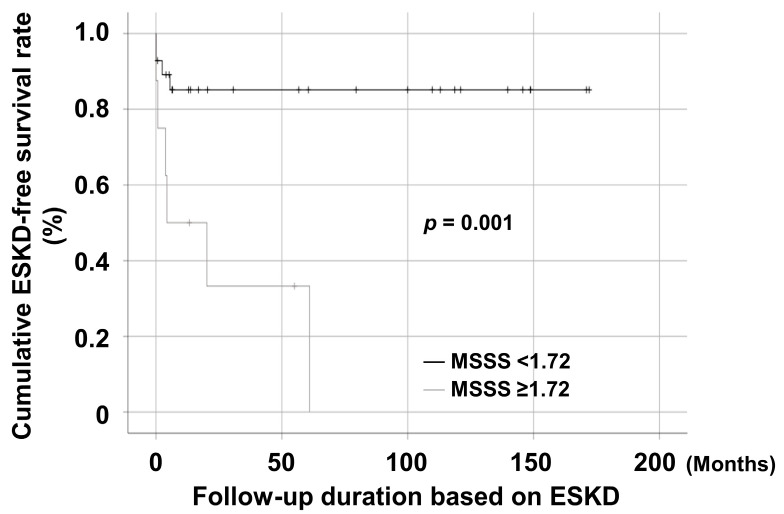
Patients with an MSSS ≥ 1.72 exhibited a significantly lower cumulative ESKD-free survival rate than those with an MSSS < 1.72. MSSS, metabolic syndrome severity score; ESKD, end-stage kidney disease.

**Table 1 jcm-10-05744-t001:** Characteristics of AAV patients with MetS.

Variables	Values
At the time of diagnosis	
Demographic data	
Age (years)	51.2 (11.2)
Sex (*n*, (%))	
Male	13 (36.1)
Female	23 (63.9)
Body mass index (kg/m^2^)	23.1 (2.9)
AAV Subtypes (*n*, (%))	
MPA	18 (50.0)
GPA	8 (22.2)
EGPA	10 (27.8)
ANCA type (*n*, (%))	
MPO-ANCA (or P-ANCA) positive	25 (69.4)
PR3-ANCA (or C-ANCA) positive	5 (13.9)
Both ANCA positive	0 (0)
ANCA positive	30 (83.3)
AAV-specific indices	
BVAS	12.5 (13.0)
FFS	1.0 (2.0)
Laboratory results	
White blood cell count (/mm^3^)	1820.0 (6170.0)
Haemoglobin (g/dL)	11.1 (3.5)
Platelet count (×1000/mm^3^)	320.0 (113.0)
Blood urea nitrogen (mg/dL)	21.7 (32.6)
Serum creatinine (mg/dL)	1.1 (2.8)
eGFR	60.8 (76.1)
Serum albumin (g/dL)	3.6 (0.8)
Acute-phase reactants	
ESR (mm/hr)	62.0 (51.0)
CRP (mg/L)	8.5 (82.4)
MSSS variables	
Waist circumference (cm)	84.3 (9.1)
Systolic blood pressure (mmHg)	130.0 (10.0)
Triglyceride (mg/dL)	146.0 (81.0)
HDL-cholesterol (mg/dL)	43.0 (17.0)
Fasting plasma glucose (mg/dL)	100.5 (38.5)
MSSS	1.1 (1.0)
During follow-up	
Poor outcomes and follow-up duration	
All-cause mortality (*n*, (%))	1 (2.8)
Follow-up duration based on all-cause mortality (months)	61.0 (100.8)
Relapse (*n*, (%))	20 (55.6)
Follow-up duration based on relapse (months)	15.9 (49.1)
ESKD (*n*, (%))	10 (27.8)
Follow-up duration based on ESKD (months)	20.4 (107.6)
CVA (*n*, (%))	2 (5.6)
Follow-up duration based on CVA (months)	58.2 (101.3)
CVD (*n*, (%))	5 (13.9)
Follow-up duration based on CVD (months)	55.5 (95.8)
Medications administered during follow-up (*n*, (%))	
Glucocorticoid	36 (100)
Cyclophosphamide	20 (55.6)
Rituximab	8 (22.2)
Mycophenolate mofetil	7 (19.4)
Azathioprine	17 (47.2)
Tacrolimus	1 (2.8)
Methotrexate	2 (5.6)

Values are expressed as a median (interquartile range, IQR) or *n* (%). AAV, ANCA-associated vasculitis; ANCA, antineutrophil cytoplasmic antibody; MetS, metabolic syndrome; MPA, microscopic polyangiitis; GPA, granulomatosis with polyangiitis; EGPA, eosinophilic GPA; MPO, myeloperoxidase; P, perinuclear; PR3, proteinase 3; C, cytoplasmic; BVAS, Birmingham vasculitis activity score; FFS, five-factor score; ESR, erythrocyte sedimentation rate; CRP, C-reactive protein; HDL, high-density lipoprotein; MSSS, metabolic syndrome severity score; ESKD, end-stage kidney disease; CVA, cerebrovascular accident; CVD, cardiovascular disease.

**Table 2 jcm-10-05744-t002:** Cox hazards model analysis of variables at the time of AAV diagnosis for ESKD occurrence during follow-up in AAV patients.

Variables	Univariable			Multivariable		
	HR	95% CI	*p* Value	HR	95% CI	*p* Value
Age	1.019	0.944, 1.099	0.635			
Male sex	0.932	0.293, 3.638	0.920			
Body mass index	0.779	0.624, 0.974	0.028			
MPA	2.266	0.585, 8.771	0.236			
GPA	2.090	0.533, 8192	0.290			
EGPA	0.027	0.000, 7.064	0.203			
MPO-ANCA (or P-ANCA) positivity	4.810	0.607, 38.140	0.137			
PR3-ANCA (or C-ANCA) positivity	0.675	0.086, 5.334	0.710			
BVAS	1.114	1.017, 1.221	0.020	1.178	0.945, 1.469	0.144
FFS	2.726	1.309, 5.677	0.007	1.030	0.258, 4.115	0.966
White blood cell count	1.000	1.000, 1.000	0.980			
Haemoglobin	0.665	0.471, 0.940	0.021	1.186	0.782, 1.800	0.422
Platelet count	0.988	0.992, 1.004	0.998			
Blood urea nitrogen	1.023	1.010, 1.035	<0.001	0.976	0.941, 1.012	0.191
Serum creatinine	2.508	1.604, 3.919	<0.001	3.713	1.560, 8.838	0.003
Serum albumin	0.555	0.198, 1555	0.263			
ESR	1.019	0998, 1.040	0.074	1.015	0.979, 1.054	0.416
CRP	1.006	0.997, 1.015	0.209			
MSSS	1.399	0.975, 2.007	0.068	1.971	1.071, 3.630	0.029

AAV, ANCA-associated vasculitis; ANCA, antineutrophil cytoplasmic antibody; ESKD, end-stage kidney disease; MPA, microscopic polyangiitis; GPA, granulomatosis with polyangiitis; EGPA, eosinophilic GPA; MPO, myeloperoxidase; P, perinuclear; PR3, proteinase 3; C, cytoplasmic; BVAS, Birmingham vasculitis activity score; FFS, five-factor score; ESR, erythrocyte sedimentation rate; CRP, C-reactive protein; MSSS, metabolic syndrome severity score.

## Data Availability

The data used to support the findings of this study are included within the article and the Appendix A.

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
