# Peer review of "Metabolic Syndrome Severity Score, Comparable to Serum Creatinine, Could Predict the Occurrence of End-Stage Kidney Disease in Patients with Antineutrophil Cytoplasmic Antibody-Associated Vasculitis"

_jcm, 2021, doi:10.3390/jcm10245744_

Round 1

Reviewer 1 Report

The authors addressed my comments. 

Reviewer 2 Report

More patients included in the present study will reinforce the idea that MSSS can predict the occurrence of end stage kidney disease.  The author has made an explanation in the revision version and may consider adding the phrase pilot study in the title or in the article. 

Reviewer 3 Report

The authors gave a good explanation of the reviewers' opinions and revised the article accordingly. Suggesting that more patients could be enrolled, with longer follow-up time, and analysis of MSSS in patients with non-metabolic syndrome will be make the study more complete.

This manuscript is a resubmission of an earlier submission. The following is a list of the peer review reports and author responses from that submission.

Round 1

Reviewer 1 Report

Metabolic syndrome is reported to be associated with more pro-inflammatory state and relapse in AAV patient as reported by Daniela R. et al. The main idea in this paper is novel and it is well written. However,  the  number of patients in this study is limited. Therefore, it is  less convincing that metabolic syndrome severity score is a good predictor for end stage renal disease in AAV patients. It is suggested that more patient included and longer duration follow up may strengthen the idea. 

Reviewer 2 Report

In the current study, Gyu et al. describe the efficacy of a metabolic syndrome severity score (MSSS) to predict end-stage kidney disease (ESKD) in 36 patients with antineutrophil cytoplasmic antibody-associated vasculitis (AAV). The results support that the MSSS can be used to predict ESKD in AAV patients.

Major comment:

  1. There are established scoring systems in AAV to predict ESKD based on histopathological findings. Therefore, MSSS must be directly compared with these scoring systems (including Berden classification and ANCA renal risk score). 
  2. In the multivariate analysis in Table 2, serum creatinine and MSSS are independent predictors for ESKD. Therefore, it should be tested if combining creatinine/MSSS could improve predictive power. 

Minor comment:

As recommended by KDIGO, the term "renal" should be replaced by "kidney", e.g. ESKD instead of ESRD. 

Reviewer 3 Report

The research is interesting and novel which showed that MetS are associated with the outcome of ANCA vasculitis. But there are something needs to improve and clarify:

1. For table 1, there should include patients with and without MetS. Since it is quite important to show that MetS is correlated with severe outcomes.

2. Why some important data were arranged into supplementary materials? Such as figure S1 and S2, it should be showed in the main parts of the manuscript since the author have a lot of discussion about it. And it is quite meaningful for the results.

3. It is a retrospective research, the number of the included patients are too small, why not include patients age over 60? Is it because after included the results are not good enough to support the results? Author did not clarify enough.

4. The determination of the target poor outcome part, the p value for ESRD is not significant.

5. Poor outcomes were only focused on ESRD is not rational. Because patients may die by other reasons before ESRD. And MetS correlated with different target organ diseases. The baseline GFR is correlated with the final outcome, when analysis it should be used as a co-variant.

6. The author did not show the median follow up time for two cohorts.